# Severe Insulin Resistance Syndromes: Clinical Spectrum and Management

**DOI:** 10.3390/ijms26125669

**Published:** 2025-06-13

**Authors:** Monika Pliszka, Leszek Szablewski

**Affiliations:** Chair and Department of General Biology and Parasitology, Medical University of Warsaw, Chałubińskiego 5, 02-004 Warsaw, Poland; monika.pliszka@wum.edu.pl

**Keywords:** insulin receptor defects, insulin receptor antibodies, lipodystrophies, congenital generalized lipodystrophies, congenital partial lipodystrophy, acquired generalized lipodystrophy, acquired partial lipodystrophy

## Abstract

Insulin resistance is a condition wherein cells fail to adequately respond to insulin. It is a prevalent medical condition associated with several diseases, such as type 2 diabetes mellitus, metabolic syndrome, hypertension, obesity, and polycystic ovary syndrome. Insulin resistance may be involved in metabolic disturbances, such as hyperglycemia, hyperinsulinemia, dyslipidemia, hyperuricemia, endothelial dysfunction, elevated inflammatory markers, and a prothrombotic state. Severe insulin resistance syndromes are a heterogeneous group of rare disorders. These disorders are characterized by profound insulin resistance, substantial metabolic abnormalities, and different clinical manifestations and complications. They may be hereditary or acquired, caused by defects in insulin action and cellular responsiveness to insulin. Severe insulin resistance syndromes may also be due to aberrations in adipose tissue function and development. The majority of these disorders are associated with an increased risk of severe complications and mortality. This review aims to summarize the current knowledge on the epidemiology, pathophysiology, complications and prognosis of severe insulin resistance syndromes, as well as to categorize these syndromes by disease process, including defects in insulin receptor, intracellular insulin signaling defects, lipodystrophies, etc.

## 1. Introduction

Insulin resistance (IR) was first described by Himsworth in 1936. He observed that elderly or obese diabetic patients were relatively insensitive to the hypoglycemic effects of insulin [1]. It is characterized by the impaired action of insulin in the body [2]. The prevalence of impaired glucose tolerance in 2021 was around 9.1% among adults aged 26–79 worldwide. It has been suggested that in 2045, this disturbance will be diagnosed in around 638 million individuals [3]. IR is defined as “a state in which a greater than a normal amount of insulin is required to elicit a quantitatively normal response” [4]. On the cellular level, IR may be defined as “the insufficient strength of insulin signaling from the insulin receptor (INSR) downstream to the final substrates of insulin action involved in various metabolic aspects of cellular function and crosstalk” [5]. To maintain normal glucose levels (euglycemia) caused by IR in peripheral muscle, adipose tissue, liver and other tissues, pancreatic β-cells increase the output of insulin, as a compensatory action, causing hyperinsulinemia [6]. There are also some cases in which organs may develop resistance to insulin, causing affected normal insulin action and cells sensitive to insulin to not respond to its levels in the body [7,8]. This disturbance may also be associated with hyperglycemia, hyperlipidemia, disturbances in glucose homeostasis, high levels of glycated hemoglobin (HbA1c), and postprandial hyperglycemia [8]. In most cases insulin resistance has no symptoms. This may be observed during the development of pre-diabetes and diabetes, when pancreatic β-cells produce more insulin, compensating for IR [9]. Several other conditions and defects are also described as associated with IR [10]. In males, the incidence IR is higher in comparison with females. This difference is caused by the protective role of estrogens. These hormones are involved in body adiposity levels, body fat distribution, glucose metabolism, and insulin sensitivity [6]. On the other hand, regarding dependence on IR, a female predominance of prevalence can be observed, as is the case for Type A IR [11]. Disturbances of insulin action may be due to several factors, such as lipotoxicity, elevated adiposity, enhanced inflammatory signaling, endoplasmic reticulum stress, mitochondrial dysfunction, increased levels of free fatty acids, and adipokines. An important role in the impairment of insulin action is played by defects in insulin signaling, including the proximal insulin signaling abnormalities, associated with mutations in the insulin receptor gene and the dysregulation of insulin action at the post-receptor level [12].

Severe insulin resistance syndromes (SIRS) are a group of rare syndromes characterized by profound insulin resistance [13]. The prevalence of these syndromes is not well documented, but it has been suggested that 0.1–0.5% of diabetic patients have severe insulin resistance syndromes [14]. These syndromes are clinically important, as well as significantly contributing to our knowledge of the mechanisms of insulin resistance. Currently, the mechanisms of insulin resistance are poorly understood, so more research is needed to elucidate the pathogenesis of IR; it is also important for the development of therapeutic strategies to treat this polygenic multifactorial condition.

This review aims to describe severe insulin resistance syndromes, pathogenesis, and therapeutic strategies.

## 2. Characteristics of Severe Insulin Resistance Syndromes

Severe insulin resistance syndrome is defined as “a severely diminished response to insulin’s biological effects, and is characterized by substantial hyperinsulinemia, and impaired glucose response to endogenous and exogenous insulin” [13]. In patients with these syndromes, hypoglycemia can be observed, as in the case of Rabson-Mendenhall syndrome, which may precede hyperglycemia [14,15]. Fasting insulin levels above 50–70 μU/mL or levels which exceed 350 μU/mL after an oral glucose tolerance test may suggest SIRS. In healthy subjects, the fasting insulin levels are below 20 μU/mL and insulin levels after an oral glucose tolerance test are below 150 μU/mL [16]. There is also an insulin sensitivity index (Si), which may suggest severe insulin resistance [17]. Si index values below 2 × 10^4^ μU/mL·min is obtained in patients with SIRS. For healthy subjects, these values are above 5 × 10^4^ μU/mL·min [18]. The Si correlates with the insulin-mediated glucose disposal rate (M), which is determined by the euglycemic hyperinsulinemic clamp [19]. Patients with severe insulin resistance have M rates below 2 mg/kg·min, while M rates in healthy subjects are above 6 mg/kg·min [16]. Intermediate M rates suggest mild to moderate insulin resistance [10].

### 2.1. Insulin Receptor and Insulin Receptor Gene

The insulin receptor gene is located on chromosome 19p13.2–13.3 and contains 22 exons, and it codes a protein of 1382 amino acids. Previously performed animal studies have revealed that only 2.4% of the total INSRs are necessary for a complete biological response to insulin [20]. Synthesized protein belongs to the Src family of tyrosine specific protein kinases [21]. The insulin receptor is a heterotetramer that contains two extracellular α chains and two transmembrane β chains. The extracellular α chains contain the insulin-binding regions; the transmembrane β chains have intrinsic tyrosine kinase activity [22]. Most mutations in the *INSR* gene are the cause of the production of a faulty insulin receptor, which cannot thence transmit signals properly. However, insulin is present in the bloodstream, impaired INSR causes have a lessened ability to extend its effects on cells. The clinical manifestations of the disease are complex, so clear diagnosis depends on gene sequencing. Currently, more than 150 mutations in the *INSR* gene have been identified, amongst which are missense mutations, nonsense mutations, deletions, insertions, and complex rearrangements [23,24]. The effects of the mutations may be divided into five categories: (1) decreased biosynthesis of INSR, (2) impaired translocation of INSR to the cell membrane, (3) decreased affinity of insulin binding, (4) inhibition of tyrosine kinase activity, and (5) accelerated degradation of INSR [25].

### 2.2. Type A Insulin Resistance Syndrome (OMIM 610549)

Type A insulin resistance syndrome (TAIRS) was first described in 1976 in three lean adolescent females with diagnosed insulin resistance and acanthosis nigricans [18]. TAIRS is rare in clinics and is estimated to affect about 1 in 100,000 people worldwide [26]. Females have more health problems, so TAIRS is diagnosed more often in females than in males. This syndrome is caused by mutations in the insulin receptor gene [14,27]. Note, this is not a consistent finding [28]. In one study, a patient was described with TAIRS due to heterozygous missense mutation in the lamin A and lamin C (*LMNA*) gene, predicting a G602S amino acid substitution in lamin A [29], which is the main component of the nuclear lamina. In the *INSR* gene, more than 60 mutations associated with TAIRS [12] have been identified.

TAIRS has a mild phenotype, and therefore clinicians generally lack a full understanding of the disease, resulting in misdiagnosis and incorrect treatment—for example, a patient with TAIRS was misdiagnosed with polycystic ovary syndrome [23]. In some patients, hypoglycemia is observed, which requires differentiation from insulinoma [24]. In patients with TAIRS, severe insulin resistance, ovarian hyperandrogenism, and acanthosis nigricans in the absence of obesity or lipoatrophy [30] are detected. Hyperglycemia is often not observed. The pancreatic β-cell compensatory response to insulin resistance may cause hypoglycemia in TAIRS patients, although this is rare [14,15]. Features of TAIRS often do not become apparent until puberty or later, and are generally not life-threatening. This syndrome is most commonly diagnosed around puberty, which is caused by the symptoms associated with ovarian dysfunction that drive the synergistic effects of gonadotropin and insulin action on the ovaries [14,15]. The major disturbances of the condition become apparent in adolescence. Menstruation in many females does not begin by the age of 16 (primary amenorrhea), and their periods may be light and irregular (oligomenorrhea). These women develop cysts on the ovaries and excessive body hair growth (hirsutism). In affected males, the features of TAIRS are more subtle, while in some affected males the only sign is low blood glucose levels. In some patients, acanthosis nigricans may be observed. In many cases, males with TAIRS come to medical attention when they develop diabetes mellitus later, in adulthood.

TAIRS is an autosomal dominant or recessive genetic disorder caused by *INSR* gene mutations [24]. Heterozygous and, in some cases, homozygous insulin receptor gene mutations impair signals of the insulin gene and signal transduction [31,32,33]. The identified heterozygous mutations caused decreased tyrosine phosphorylation in the β-subunit after insulin binding [34]. A cause of TAIRS was described as a heterozygous mutation in the *INSR* gene c.3602G>A (p.Arg1201Gln) in a 14-year-old girl who had diabetes, but without common clinical features [35]. Other cases described a patient with type 2 diabetes mellitus with heterozygous missense mutation in the *INSR* gene, c.3472C>T (p.Arg1158Trp). This mutation was also found in the father of a patient proband. In the patient proband and in his father, high insulin and C-peptide release after glucose stimulation [24] was detected. A novel mutation in exon 19 of the *INSR* gene in an adolescent girl was described also [36]. A further next-generation sequence analysis revealed deletion in c.3486_3503delGAGAAACTGCATGGTCGC/p.Arg1163_Ala1168, and acanthosis nigricans, hypertrichosis, fasting hyperglycemia, fasting and postprandial hyperinsulinemia and biochemical hyperandrogenemia were detected. The same mutation was described in the patient’s mother, who had a milder clinical phenotype [36]. Other investigations of adolescent women showed hirsutism, elevated levels of insulin and testosterone, and multiple cystic changes in the bilateral ovaries. The previous diagnosis was polycystic ovary syndrome, but additional tests revealed that this diagnosis was incorrect, the correct diagnosis being TAIRS, in this case TAIRS being due to the c.3601C>T heterozygous missense mutation in the *INSR* gene. This mutation was also detected in the proband’s father and two brothers [23]. Heterozygous missense mutation-caused TAIRS was also described in a Caucasian girl with normal BMI, but severe hirsutism, acanthosis nigricans, clitorial hypertrophy, enlarged polycystic ovaries, hyperinsulinemia, and hyperandrogenism were also seen. A mutation detected in the tyrosine kinase domain of the *INSR* gene (p.Leu1150Pro) of the proband was also detected in the patient’s proband mother and brother [30]. Other heterogenous mutations have also been described in the *INSR* gene, such as c.3614C>T [37,38], p.Val1086del heterozygous mutation in the proband and his mother and grandfather [39], and c.3670G>A (p.Val1224Met) that had no effect on the total protein expression and phosphorylation of INSR. Based on the results obtained, the authors suggest that the variant p.Val1224Met may be tolerated and is not associated with severe insulin resistance [38]. Patients with TAIRS can live beyond middle age [40].

### 2.3. Type C Insulin Resistance Syndrome

Type C insulin resistance syndrome, associated with a distinctive clinical phenotype of hyperandrogenism, insulin resistance, and acanthosis nigricans (HAIR-AN), was first described in 1968 by Brown and Winkelmann [41]. Researchers described 90 cases of acanthosis nigricans, 2 of which were women with signs of Stein–Leventhal syndrome. In these women amenorrhea and hirsutism were seen; one women was very obese, while the other was not so obese [41]. It has been suggested that one patient had type C insulin resistance syndrome, while the other had type A insulin resistance syndrome [42]. Some patients with HAIR-AN may have markedly elevated levels of fasting insulin (>80 μU/mL) [43]. The degree of obesity in patients with type C insulin resistance syndrome plays an important role in determining the degree of insulin resistance [44,45]. In these patients are observed increased basal rates of hepatic glucose production and peripheral insulin resistance [46]. Patients with HAIR-AN are at a greater risk of metabolic dysfunction and diabetes mellitus [42].

It is suggested that HAIR-AN affects 2–3% of androgen-excess patients [47,48]. On the other hand, sometimes, it is the HAIR-AN syndrome as differentiated from polycystic ovary syndrome (PCOS) that is often associated with insulin resistance. It should be noted that it has not yet been fully clarified whether this syndrome is distinct from type A and b syndromes or PCOS [16]. However, according to the suggestions of other researchers, HAIR-AN is a variant of type A insulin resistance syndrome [11]. Affected women with normal *INSR* genes [49], however, presented observations that may suggest insulin receptor defects [50]. It is also suggested that HAIR-AN is caused by reduced insulin receptor kinase activities [11]. The results obtained in experiments have revealed a normal binding of insulin to monocytes and fibroblasts [51,52], an observation that may suggest the postreceptor defects of insulin signaling. Patients with type C insulin resistance syndrome are often treated as other patients with PCOSs, with lifestyle modifications, oral contraceptives, antiandrogens, and insulin sensitizers [53,54]. The inheritance of HAIR-AN is not known [11]. On the other hand, based on the observations undertaken, it is suggested that type C insulin resistance and short stature are dominantly inherited autosomally [55]. The authors described a 17-year old girl with insulin-resistant diabetes mellitus, acanthosis nigricans, hirsutism, and short stature. Insulin binding to erythrocytes was normal, suggesting a postreceptor defect in the insulin signaling pathway. The insulin-resistant diabetes mellitus and short stature were also detected in her mother, maternal grandfather, and uncle. Interestingly, these individuals were reportedly affected with the same phenotype. In her father, they detected impaired glucose tolerance, but not hyperinsulinemia, and two of her sisters had normal glucose tolerance. The degree of insulin binding to the erythrocytes of her parents was within the normal range. Based on these results, researchers suggest that HAIR-AN and short stature are dominantly inherited autosomally [55]. The prognosis of HAIR-AN is generally not life-threatening [11].

### 2.4. Type B Insulin Resistance Syndrome (ORPHA 2298)

Type B insulin resistance syndrome (TBIRS) was first described in 1975 by Flier et al. [56]. It is a very rare autoimmune disorder caused by polyclonal autoantibodies, usually IgG against INSR [13]. TBIRS most commonly causes extreme insulin resistance with severe hyperglycemia and weight loss; however, hypoglycemia may be visible [57,58,59,60], though its prevalence is unknown [11]. TBIRS is most common in African Americans, but it has also been reported in whites and Asians. Only a few cases have been reported in Hispanic or English literature, or from other so-called developed countries [61,62,63]. In Japan, the proportion of newly diagnosed patients that test positive for autoantibodies against INSR is 22% a year [64], a result that also indicates that TBIRS is relatively rare [65].

Previously performed experiments revealed that the autoantibodies may act biphasically, inducing hypoglycemia in the first (acute) phase, while ultimately causing hyperglycemia [13]. It is suggested that the effects of autoantibodies are associated with their concentrations. High autoantibody levels inhibit the INSR (antagonistic role), causing insulin resistance and hyperglycemia, while low levels of these autoantibodies partially agonize insulin-mimetic activity, causing hypoglycemia [66,67,68]. Therefore, in patients, it is detected that profound insulin resistance and hyperglycemia occur, though hypoglycemia sometimes less so [13]. On the other hand, there are observations that suggest that about 76% of cases of TBIRS are complicated with hypoglycemia [64], which together with hyperglycemia is an important clinical feature [65]. Hypoglycemia has also been described as the initial presentation, most commonly in white human beings [63], and associated with high mortality [57]. Results obtained in other observations suggest that hypoglycemia is detected in approximately 43% at some stage during the course of the disease. It was also found that approximately half of individuals have recently, or had previously, undergone insulin therapy. In total, 22% of patients were diagnosed with isolated hypoglycemia [62]. Isolated hypoglycemia in the presence of autoantibodies against INSR is exceptionally rare, and currently considered part of type B resistance syndrome [59]. TBIRS may be associated with type 1 diabetes mellitus [69], as well as with type 2 diabetes mellitus [70]. Usually, TBIRS is associated with other autoimmune diseases. The most common autoimmune disorder in patients with TBIRS was systemic lupus erythematosus (SLE), in up to 35.3% of cases [62], or in 20% of patients [65]. TBIRS may be also associated with other autoimmune diseases, such as Sjögren syndrome, systemic scleroderma, mixed connective tissue disease, autoimmune thyroid disease, autoimmune thrombocytopenia, rheumatoid arthritis, and psoriasis [18,57,65,71,72,73]. Malignancies such as Hodgkin’s disease and myeloma were also diagnosed as TBIRS [62,74].

TBIRS is difficult to diagnose, because it may present as hyperglycemia or hypoglycemia, which are difficult to control. The basis of the diagnosis may be high fasting insulin levels, hyperadiponectinemia, or normal or low triglycerides [57,62,75]. Diagnosis is based on the finding of positive anti-insulin receptor antibodies. It is suggested that patients who require doses greater than 3 U/kg/day and have concomitant autoimmune diseases should be examined for TBIRS [62]. TBIRS has different clinical presentations, but the most common is severe hyperglycemia. These patients require an average daily insulin dose of approximately 5000 units, ranging from 200 to 25 000 IU/day [57]. TBIRS is typically diagnosed predominantly in young to middle-aged women (mean age 44 years, in a range of 15–69 years), in association with other autoimmune diseases [42,57]. The principal clinical features include acanthosis nigricans, mainly in the periocular, perioral and labial regions, and in the axillae, neck and groin [76], plus weight loss, high testosterone levels [57], low triglyceride levels [77], and high levels of adiponectin [75]. In premenopausal women with TBIRS, hyperandrogenism is common, associated with enlarged cystic ovaries [57]. In women with TBIRS, amenorrhea and hirsutism may also be observed, and some may develop virilization. The above-mentioned features of ovarian hyperandrogenism may depend on the degree and duration of TBIRS [42]. Remission may be described as the amelioration of glucose abnormality, or the normalization of fasting insulin concentration and the disappearance of anti-insulin receptor antibodies [67]. Remission may occur spontaneously or may be caused by a specific intervention. In several case reports, the complete remission of TBIRS after immunotherapy has been observed [70,78]. Only short-term remission data were presented, while long-term data are few in number [11,70]. Observations performed by the United States National Institute of Health revealed spontaneous remission in 33% of patients. Remission caused by therapy was reported in another 33% of patients, and the remaining 33% did not achieve remission despite a long period of follow-up. The time taken to achieve remission is highly variable, ranging between 6 and 27 months [79]. TBIRS manifests itself in adulthood, and mortality has been reported to be more than 50% within 10 years of diagnosis [57,79].

### 2.5. Donohue Syndrome (OMIM 246200)

Donohue syndrome (DS), also referred to as leprechaunism, was first described in 1954 by Donohue and Uchida [80]. This is a very rare congenital syndrome of extreme insulin resistance, with a prevalence of approximately one in every million live births [81,82]. In medical literature, only a few cases of DS have been documented [83]. The clinical presentation of DS manifests with impaired glucose homeostasis and severe growth impairment. Intrauterine and postnatal growth restriction with a large number of other dysmorphic features are also detected [84]. In patients with DS, features are observed such as hyperinsulinemia, fasting hypoglycemia, postprandial hyperglycemia, lipoatrophy, glucose intolerance or overt diabetes mellitus, β-cell hypertrophy, severe prenatal growth restriction, postnatal growth failure, hypotonia, and developmental delay. In affected patients, characteristic facial features are due to dysmorphia, including proptosis, infraorbital folds, low-set and posteriorly rotated ears, hypertelorism of the orbits, thick vermilion of the upper and lower lips and gingival hyperatrophy, and a flattened nasal bridge with a lack of subcutaneous fat. Hyperatrophy of the internal organs, including cardiomegaly, hepatosplenomegaly, and hyperatrophy of the kidneys and ovaries, is also observed. Hypertrichosis, acanthosis nigricans, polycystic ovaries, and breast enlargement in female infants are also observed. Affected female patients may have clitoromegly and cystic ovaries, and in affected male patients, penile enlargement may also be observed. Mental retardation is also observed [42,83,85,86].

DS is caused by an autosomal recessive mutation in the insulin receptor gene [83,86]. These mutations cause the disrupted binding of insulin to INSR and intracellular insulin signaling pathway [87]. As mentioned above, only a few cases of DS have been described. Different mutations in the *INSR* gene and their effects on protein are detected; for example, for mutation c.320C>G, a p.Thr107Arg homozygous missense mutation was found in exon 2 [86]. This mutation was detected in male infants with congenital insulin resistance. Intrauterine growth retardation, transient hypoglycemia, pneumonia, urinary tract infection, and heart defects were diagnosed after birth; hyperglycemia was also observed [86], and another case describes a male newborn [88]. A genetic study revealed two compound heterozygous mutations in *INSR* gene p.R1026X/D1179GfsX8. The first mutation was described earlier by Kadowaki et al. [89]. In one infant, intrauterine hypoglycemia, growth delay, an elfin face with low-set ears, and a flat nasal root, macrostomia, acanthosis nigricans, hypertrichosis, pachyderma with no fatty layer in the chest and arms, and a very developed scrotum were detected. The infant experienced frequent respiratory infections and died at 19 months of age from a serious respiratory tract infection [88]. In newborns there are also described other mutations in the *INSR* gene and insulin receptors, such as c.742_751del (p.Thr250del) in the α-subunit, which reduces the mature insulin receptor and impairs its function [38]. In other female newborns, a lack of subcutaneous fat tissue, hypertrichosis, gingival hypertrophy, hyperandrogenism, postprandial hyperglycemia, and other features associated with DS were detected [87]. *INSR* gene analysis revealed a compound heterozygous mutation p.R813 (c.2437C>T), which was also detected in her father, and p.777–790delVAAFPNTSSTSVPT was also shown in her mother. The parents were heterozygous for these mutations. The newborn died after 75 days of severe heart failure and pneumonia [87]. This was also described in another case of Donohue syndrome [90]. A homozygous missense mutation p.Leu795Pro was found in β-subunit of INSR. A male newborn had several features characteristic for DS; he died at the age of 6 months, due to a fulminant pulmonary infection [90].

Common causes of death are recurrent bacterial infections, including pneumonia and urinary tract infection [86]. Some patients with DS die during their first two years of life [91]; however, death usually occurs within the first year [85].

### 2.6. Rabson–Mendenhall Syndrome (OMIM 262190)

Rabson–Mendenhall syndrome (RMS) was first described in 1956 by Rabson and Mendenhall in three siblings with extreme insulin-resistant diabetes, acanthosis nigricans, thick rapidly growing scalp hairs, phallic enlargement, precocious pseudopuberty, markedly thickened nails, dental abnormality, and pineal hyperplasia [92]. Its exact prevalence has not been assessed [93], but recent data on congenital insulin resistance linked to insulin receptor gene mutations that include RMS and DS set the incidence at ≈1 in 1,200,000–1,300,000 live births [94]. Its prevalence in less than 1 in 1,000,000 live births has also been suggested [95]. Rabson–Mendenhall syndrome is related to Donohue syndrome, and is characterized by severe insulin resistance, although not as severe as that of DS, in addition to acanthosis nigricans and hyperplasia of the pineal gland, as well as several dysmorphic features. In terms of severity, RMS is intermediate amongst DS and TAIRS [95]. In affected individuals, growth retardation during pregnancy and after delivery is diagnosed, often causing short stature later in childhood. There are also other features, such as adipose tissue hypotrophy, causing a lack of subcutaneous fat, muscle atrophy, dental abnormalities, hirsutism, polycystic ovaries, enlarged nipples and genitalia (clitoromegaly in females and enlarged phallus in males), nephromegaly, precocious pseudopuberty, abdominal distension, hyperglycemia and hyperinsulinemia [93,95,96,97,98,99]. Facial features here may be milder than those seen in DS [85]. RMS differs from DS in the presence of premature and dysplastic dentition, coarse facial features, and pineal hyperplasia [100]. RMS occurs due to mutations in the *INSR* gene, causing decreased insulin binding to its receptor. It is an autosomal recessive genetic disorder [101]. Several different mutations in the *INSR* gene have been detected that are associated with RMS. Based on PubMed Embasa, the China National Knowledge Infrastructure and Wanfang databases, clinical analyses have been performed on gene mutations in adult patients with RMS [97]. Investigations of gene mutations in 42 patients with RMS showed that 20 patients (47.62%) had only one mutation, while 22 patients (52.38%) with RMS had two or more mutations. A total of 55 distinct *INSR* gene mutations, such as missense (40 mutations, 72.72%), deletion (6 mutations, 10.91%), insertion (1 mutation, 1.82%), and nonsense (1 mutation, 1.82%) were found. The most prevalent was p.Glu238Lys mutation. The clinical characteristics of these patients revealed hyperinsulinemia (100%), non-obesity (100%) or underweight (84.21%), acanthosis nigricans (69.05%), growth retardation (59.52%), dental anomalies (54.76%), and hirsutism (40.48%) [97]. In one Paraguayan patient, a 5-year-old girl, the first case of RMS was reported [102]. The affected girl had hypertrichosis, acanthosis nigricans, nephrocalcinosis, an acromegalic face, ogival palate, hyperkeratosis, rough skin, and rough hair. When she was 8 months old, she visited the clinic for a consultation associated with suspected early puberty. Genetic testing revealed two pathogenic variants in exons 2 and 18 of the *INSR* gene, c.332G>T (p.Gly111Val) and c.3485C>T (p.Ala1162Val), combined as heterozygous [102]. Another case described a 15-year-old female patient with poorly controlled diabetes [103]. At the age of just 50 days, she was presenting with diabetic ketoacidosis (DKA), and was diagnosed with RMS. She had four episodes of DKA associated with acute infection–right lobar pneumonia, cellulitis of the abdominal wall, acute gastroenteritis, and lipohypertrophy. A genetic study revealed a homozygous mutation for R141W in the *INSR* gene. A strong familial history of RMS was found in her mother’s cousins, with one affected and two carriers [103]. RMS was also detected in a brother–sister pair [104]. The boy was referred with high plasma insulin and dysmorphic features associated with RMS, such as coarse facial features with globular nose, full lips and furrowed tongue, and hyperkeratotic skin with hypertrichosis. His sister had diagnosed diabetes, and presented with similar dysmorphic features, such as acanthosis nigricans, dental abnormalities, and bilateral nephrocalcinosis. Both affected children were found to be homozygous for the p.Arg141Trp missense variant in the α-subunit of INSR [104]. RMS was also described in a male of 45 years of age, who was diagnosed with RMS at 11 years [105]. In the patient, characteristic features were detected such as acanthosis nigricans, macroglosia, dystrophic nails, precocious puberty, short stature, and severe insulin resistance. Molecular analysis revealed that the patient had in one allele a nonsense mutation substituting the opal chain termination codon (TAG) for arginine (CGA) at codon 1000. This mutation markedly decreases the levels of INSR mRNA transcribed. In the other allele, a missense mutation was detected that substitutes lysine (AAA) for asparagine (AAC) at codon 15. This mutation impairs the post-translational modification and transport of INSR to the cell surface, and also causes a 5-fold decrease in the affinity of insulin binding [105]. Similar pathologies were also detected in other patients with RMS [106,107,108].

Patients with RMS survive early childhood, but they have a significantly reduced life expectancy and may die during adolescence or early adulthood [103,104]. Patients with RMS usually survive beyond a single year, but they develop several pathologies early in life, and live into their second or third decade [95,98], although death usually occurs in the second decade, and most patients survive only up to 15 years of age [95]; however, one case study from Spain reported long-term survival in a patient with Rabson–Mendenhall syndrome [105]. Complications of long-lasting hyperglycemia and complications associated with diabetes mellitus, such as diabetic ketoacidosis, are the most common causes of death [85,95].

Severe insulin resistance syndromes due to mutations in the insulin receptor gene and/or disturbances of the intracellular insulin pathway are summarized in Table 1.

### 2.7. Lipodystrophies

Lipodystrophic syndromes may also be associated with both insulin resistance and ovarian hyperandrogenism or dysfunction. These syndromes are primarily characterized by the complete or partial loss of adipose tissue and the depletion of lipid storage capacity [109,110].

Lipodystrophies are a group of very rare disorders. Their worldwide prevalence, if excluding the HIV-related form, was estimated to be 3.07 cases per million (0.23 cases/million of generalized lipodystrophy, and 2.84 cases/million of partial lipodystrophy). Based on results from literature searches, the prevalence of lipodystrophy syndromes in Europe has been predicted to be 0.96 and 1.67 cases/million for generalized and partial lipodystrophies, respectively [111]. Lipodystrophy that occurs in patients with HIV infection is the most prevalent form. The most prevalent type of lipodystrophy is an acquired form that is detected in HIV-infected patients treated with highly active antiretroviral therapy (HAART). The investigations performed revealed that up to 40–70% of patients on HAART have diagnosed HIV-associated lipodystrophy syndrome (HALS) [112,113].

These diseases may be due to genetic mutations, autoimmune processes or medication effects [114]. Lipodystrophies may be classified according to the distribution of fat loss. The fat loss can be seen across the whole body (generalized lipodystrophy, GL) or in specific areas (partial lipodystrophy, PL). Both forms of lipodystrophy syndromes are congenital or acquired, leading to four major categories: congenital generalized lipodystrophy (CGL), familial partial lipodystrophy (FPLD), acquired generalized lipodystrophy (AGL), and acquired partial lipodystrophy (APL) [13,42,115,116,117]. In many affected individuals, especially those with GL, low levels of leptin are observed, which plays an important role in energy homeostasis, the metabolism of the lipid and the action of insulin [118]. Some have detected metabolic disorders and especially insulin resistance, severe hyperlipidemia such as hypertriglyceridemia, as well as diabetes mellitus, progressive hepatic diseases and an increased metabolic rate [119,120]. Affected female patients may develop clinical signs of ovarian hyperandrogenism [121,122].

#### 2.7.1. Congenital Generalized Lipodystrophies

Congenital generalized lipodystrophy (CGL, OMIM 269700), also known as Berardinelli–Seip syndrome, was described initially in 1954 by Berardinelli, and five years later by Seip [123,124]. It is a rare but clinically important disorder. In this lipodystrophy, an almost complete lack of adipose tissue, both at birth and in early childhood, is observed [125]. CGL is an autosomal recessive disease [126]. Approximately 300 patients from different ethnic groups have been described, with the highest frequency reported in Brazil [13,127]. CGL is most prevalent among children of consanguineous parents [128]. The syndrome is diagnosed soon after birth. Affected children have voracious appetites and accelerated linear growth rates. CGL patients are also reported to have diabetes mellitus, hypertriglyceridemia, hepatic steatosis, cirrhosis, acromegaloid features, acanthosis nigricans, cardiomyopathy, mild mental retardation, advanced bone age, cervical spine instability, and muscular weakness [127], as well as complications associated with diabetes, such as nephropathy and retinopathy [129]. In women with CGL, reproductive function is affected, and clitoromegaly, hirsutism, amenorrhea, and polycystic ovaries are detected. Only a few females had successful pregnancies, but most males were reported to be fertile [127,130]. The levels of hormones produced by adipose tissue, such as leptin and adiponectin, are typically low [131]. Observations made in Brazil revealed that the average age of death of CGL patients was 27.1 ± 12.4 years. Most patients had CGL2 and the most common causes of death were infections, such as pneumonia, and liver complications, such as cirrhosis [132].

To date, at least four molecularly distinct CGL types have been identified, known as congenital generalized lipodystrophy type 1 (CGL1), type 2 (CGL2), type 3 (CGL3), and type 4 (CGL4) [126]. CGL1 and CGL2 are responsible for 95% of reported cases [13].

Type 1 CGL (CGL1, OMIM 608594)

CGL1 is an autosomal recessive disorder, caused by mutations in the *AGPAT2* gene. *AGPAT2,* located on chromosome 9q34, codes the enzyme acyltransferase 1-acylglycerol-3-phosphate O-acyltransferase 2 (AGPAT2), which is a key enzyme in the synthesis of triglyceride. It catalyzes the acylation of lysophosphatidic acid to form phosphatidic acid, which is intermediate in the biosynthesis of triglycerides and glycerophospholipids [133]. The deficiency of AGPAT2 causes impaired signaling pathways in PI3K/AKT and PPARγ, resulting in impaired adipogenesis and reduced levels of stored triglycerides inside the adipose tissue [134]. Lysophosphatidic acid (LPA) may be involved in inflammation and fibrosis in the adipose tissue, causing a loss of adipose tissue. Its accumulation in the liver may trigger the progress of metabolic dysfunction-associated fatty liver disease (MAFLD) to metabolic dysfunction-associated steatohepatitis (MASH) [135]. It has been described in at least 150 cases and 42 mutations in the *AGPAT2* gene [136,137]. In patients with CGL1, normal adipose tissue deposition is limited to mechanical fat (orbital regions, palms, soles, and joints), the mouth and tongue, and the scalp and perineum, while metabolically important adipose tissue (subcutaneous, bone marrow, intraabdominal, intermuscular, and intrathoracic depts) is markedly reduced [138,139].

Type 2 CGL (CGL2, OMIM 269700)

CG21 is an autosomal recessive disorder caused by mutations in the *BSCL2* gene, localized on chromosome 11q13. It codes the protein seipin, an integral endoplasmic reticulum membrane protein [140]. At least 36 mutations of *BSCL2* gene and 167 patients with CGL2 have been described [136]. Seipin is involved in lipid droplet formation and the differentiation of adipocyte [141,142]. CGL2, in comparison with CGL1, is more severe, with a higher incidence of premature death. This syndrome also features a lower prevalence of partial and/or delayed onset of lipodystrophy. Patients with CGL2 are born without any adipose tissue, and show a loss of both metabolically active fat and mechanical fat [127]. The prevalence of intellectual impairment in patients with CGL2 is higher compared to patients with CGL1 [143,144]. These patients also have a higher prevalence of cardiomyopathy in comparison to patients with CGL1 [145]. In the early childhood of CGL2 patients, hypertriglyceridemia and hepatic steatosis can be detected, and hepatic involvement may be more severe in CGL2 as compared to other types [146,147].

Type 3 CGL (CGL3, OMIM 612526)

CGL3 is caused by homozygous nonsense mutations in the *CAV1* gene, located on chromosome 7q31. This syndrome is inherited in an autosomal recessive manner. *CAV1* encodes the protein caveolin-1. Caveolin-1 is an integral part of the plasma membrane microdomains, the caveolae. It is involved in cell migration, polarization and proliferation [148]. This protein is also associated with the PKA-mediated phosphorylation of perilipin, which regulates lipolysis [149]. Impaired proteins lead to lipodystrophy, caused by disordered lipid handling, lipid droplet formation, and/or adipocyte differentiation. The impaired function of caveolin-1 decreases lipid accumulation, resulting in white adipose atrophy [149]. Patients with CGL3 have an intermediate phenotype between CGL1 and CGL2. CGL3 differs from other types of CGL, due to the presence of unique features, such as preserved bone marrow fat, vitamin D resistance, hypocalcemia, hypomagnesemia, and decreased bone density [116]. In CGL3, mechanical fat is preserved and intellectual disability is absent. Heterozygous null *CAV1* pathogenic variants were also identified de novo in two patients from Europe, with generalized fat loss, thin mottled skin, and progeroid features at birth [150], with associated lipodystrophy heterozygous *CAV1* frameshift mutations also reported. In these patients, features such as congenital cataracts and cerebellar progressive ataxia were detected [151,152]. Mutation His79Glnfs*3, identified in four consanguineous patients diagnosed with CGL3, was also described. In these patients, typical features were observed, as well as additional features, such as esophageal achalasia in two patients, while another had atypical retinitis pigmentosa [153].

Type 4 CGL (CGL4, OMIM 613327)

CGL4 is a rare, autosomal recessive disorder. It is caused by mutations in the *PTRF* gene, which encodes cavin. Cavin is a polymerase 1 and transcript release factor that regulates the expression of caveolae 1 and 3, as well as stabilizing and assembling the membrane structure [154]. Cavin is a peripheral membrane protein and a component of caveolae. Cavin regulates the differentiation of adipocyte and the expandability of adipose tissue [155,156]. The majority of CGL4 patients have no mutations in the *CAVIN1/PTRF* gene; however, a new homozygous mutation, c.21T>A, p.Tyr7Ter, was described in two pediatric siblings, whose clinical manifestations were characteristic for CGL4 [157]. In infants with CGL4, progressive body fat loss may be detected. This syndrome is characterized by myopathy, pyloric stenosis, gastrointestinal dysmotility, QT interval prolongation with exercise-induced ventricular tachycardia, atlantoaxial instability, scoliosis, and sudden death [158,159,160]. The congenital generalized lipodystrophies are summarized in Table 2.

#### 2.7.2. Congenital Partial Lipodystrophies

Congenital partial lipodystrophies are genetic syndromes characterized by regional lipoatrophy [13]. Partial lipoatrophy may be categorized into inherited (familial partial lipodystrophy, FPLD) and acquired forms (acquired partial lipodystrophy, APL). Both patients with FPLD and APL start losing fat at some point during their life. Patients with FPLD are observed to experience variable losses of adipose tissue during adulthood. The fat loss usually occurs earlier in girls, and is seen during late childhood or puberty [161]. These syndromes are also associated with metabolic complications, and, in some cases, cardiomyopathy, conduction disturbances, and congestive heart failure may be detected. Some clinical characteristics and metabolic disturbances, such as insulin resistance, hyperglycemia, diabetes mellitus, acanthosis nigricans, hypertriglyceridemia, hepatic steatosis, hyperandrogenemia, hirsutism, polycystic ovary syndrome, reproductive dysfunction, and osteoporosis, are observed [161]. The majority of FPLD syndromes are inherited in an autosomal-dominant manner, and revealed varying degrees of subcutaneous fat loss [129,162]. Partial lipodystrophy syndromes were once considered rare [163]. On the other hand, there are observations that suggest a prevalence of these syndromes of approximately 1 in 20,000 [164].

The six molecularly distinct types of FPLD have been defined.

Type 1 FPLD (FPLD1, OMIM 608600)

FPLD1 or Köbberling-type lipodystrophy is predominantly diagnosed in females [165]. The genetic basis of this lipodystrophy syndrome is unknown, and it is suggested that a polygenic inheritance pattern exists in some individuals [166,167]. It may also be caused by environmental factors [168]. In patients with this syndrome, a loss of adipose tissue in the extremities and normal adipose tissue elsewhere is observed. In the face and neck, fat may be normal or slightly increased, and a common binding includes truncal obesity. In affected individuals, the formation of a palpable “ledge” between the normal and lipodystrophic areas occurs [165]. The age of onset and specific features are not completely defined. Most patients have diabetes mellitus and hypertriglyceridemia, and this may cause acute pancreatitis [169].

Type 2 FPLD (FPLD2, OMIM 151660)

FPLD2, or Dunningan lipodystrophy, is caused by autosomal dominant mutations in the *LMNA* gene, located on 1q21-22. This syndrome is the most common type. More than 500 cases have been reported [161]. The *LMNA* gene codes nuclear lamina proteins, lamin A and lamin C. These proteins, localized in a nuclear envelope, are involved in the organization of the nuclear architecture. Pathogenic variants of the *LMNA* gene are caused by missense mutations. Impaired proteins may disrupt interaction with chromatin or other nuclear proteins. These disturbances cause apoptosis and premature death of the adipocyte [170]. The accumulation of prelamin A may impair adipogenesis. The disruption of adipogenesis caused by the accumulation of prelamin A may be due to its interference with adipocyte transcription factors and regulators, such as sterol response element binding protein 1 (SREBP1) and peroxisome proliferator-activated receptor-γ (PPARγ) [170,171,172]. FPLD2 is usually associated with the gradual onset of subcutaneous fat loss from the extremities during puberty. It is also associated with increased musculature. In affected patients, hypertrophy of type 1 and type 2 muscle fibers is observed [173]. In these patients, the development of excess supraclavicular fat and round faces in life may be detected. They also may have acanthosis nigrians, hirsutism, and hyperandrogenism. This syndrome is relatively easier to recognize in females, and they have more severe metabolic complications [174]. Most affected patients develop diabetes mellitus in their twenties and thirties, and have a high risk of cardiovascular diseases, which develop at younger ages [175]. Among patients with FPLD2, there is a phenotypic heterogeneity. These differences depend on the site of mutation in the *LMNA* gene. For example, mutations in exon 11 affect only lamin A, those in exon 6 cause facial fat loss, and mutations in exon 1 are associated with severe cardiac disease that requires cardiac transplant at an early age [175,176,177,178].

Type 3 FPLD (FPLD3, OMIM 604367)

FPLD3 is due to autosomally dominant mutations in the *PPARG* gene, located on chromosome 2p25, which codes peroxisome proliferator-activated receptor-gamma. It is involved in adipocyte differentiation and its function. FPLD3 is the second most common type of FPLD (approximately 20 families were described). It reveals metabolic complications at a similar rate and severity to patients with FPLD2 [179,180,181]. Patients with this syndrome usually show milder fat loss. In affected individuals, lipoatrophy appears in adulthood. In the face and neck, no accumulation of adipose tissue is observed. FPLD3 patients have a notably greater prevalence of hypertriglyceridemia and diabetes mellitus [182]. It was found that a heterozygous mutation may attenuate the expression of the *PPARG* gene, or may interfere directly with normal gene function (dominant negative), inhibiting adipocyte differentiation [13].

Type 4 FPLD (FPLD4, OMIM 613877)

FPLD4 occurs due to autosomally dominant mutations in the *PLIN1* gene that codes perilipin-1 [161]. This syndrome has been described in four families [13]. Perilipn-1 is a phosphoprotein in adipocytes. It is an essential component of the lipid droplet coat protein [183]. This protein is involved in lipid storage and lipolysis by regulating hormone-sensitive lipase (HSL) and adipose triglyceride lipase (ATGL). These enzymes cause the hydrolysis of diacylglycerol and triacylglycerol into monoacylglycerol and fatty acids [184]. Although frameshift mutations in the *PLIN1* gene cause partial lipodystrophy, there are observations that suggest that null variants in this gene are not associated with lipodystrophy [185]. In FPLD4 patients, the loss of adipose tissue is observed, especially in the lower limb and femorogluteal depot. This syndrome may cause severe insulin resistance, diabetes mellitus, hypertriglyceridemia, and hepatic steatosis [186,187,188].

Type 5 FPLD (FPLD5, OMIM 615238)

FPLD5 is an autosomal recessive syndrome caused by mutations in the *CIDEC* gene, located on chromosome 3p25.3. This gene codes Cell Death Inducing DFFA like Effector C protein. This protein, which is expressed in the lipid droplets, plays a role in the differentiation of adipocyte and is involved in the regulation of lipid and glucose metabolisms. Mutations in the *CIDEC* gene impair adipocyte differentiation and lead to the inability of lipid droplets to store fat [189,190]. In affected patients, partial lipodystrophy, acanthosis nigricans, severe insulin resistance leading to diabetes mellitus, and hepatic steatosis are detected [189].

Type 6 FPLD (FPLD6, OMIM 615980)

FPLD6 is an autosomal recessive syndrome caused by mutations in the *LIPE* gene, which codes lipase E of the hormone-sensitive type. This hormone is associated with adipocyte function, and the homeostasis of lipid and glucose. It regulates lipolysis from adipocytes. Pathological mutations in the *LIPE* gene impair lipolysis, which may cause lipomatosis and partial fat loss, associated with hypertriglyceridemia, hepatic steatosis, insulin resistant diabetes, and inflammation [191]. In affected patients, late-onset partial fat loss from the lower extremities, multiple symmetric lipomatosis, and progressive distal symmetric myopathy are observed [191,192].

AKT2-linked lipodystrophy

FPLD may also be caused by autosomal dominant mutations in the *AKT2* gene (protein kinase B), a serine/threonine-protein kinase. This kinase is involved in several processes, such as cell signaling, cell growth, glycogen synthesis, and insulin-stimulated glucose transport. Patients with *AKT2* mutations have impaired insulin signaling and adipocyte differentiation [193].

CAV1-associated lipodystrophy (OMIM 606721)

A pathogenic variant in *CAV1*, inherited in an autosomally dominant manner [117], is rare sporadic partial lipodystrophy [152]. CAV1 (caveolin-1) is a membrane protein involved in cell metabolism, cell migration, and cell signaling. There are only two cases reported with different frameshift *CAV1* variants. In both cases, patients suffer partial lipodystrophy with subcutaneous fat loss in the face and upper body, micrognathia, congenital cataracts, diabetes mellitus, hypertriglyceridemia, and recurrent pancreatitis. One patient had abnormal neurologic findings [152].

Mandibuloacral dysplasia

Mandibuloacral dysplasia (MAD) is an extremely rare autosomal recessive syndrome. It has been identified in approximately 40 cases [194,195]. In patients with MAD, features are observed such as craniofacial, skeletal and cutaneous abnormalities, such as mandibular and clavicular hypoplasia, birdlike face, delayed closure of the cranial sutures, acro-osteolysis, joint countractures, dental abnormalities, mottled cutaneous pigmentation, restrictive dermatopathy, skin atrophy, and alopecia. Lipoatrophy is noted in childhood or early adolescence and is more marked in affected females. Progeroid features and dysmorphic disturbances may be present at birth, and the full clinical phenotype is observed during early school years [114,196,197].

There are two distinct phenotypes of MAD. MAD type A (OMIM 248370) is caused by mutations in the *LMNA* gene that codes lamin A/C nuclear lamina proteins. Mutations in this gene cause the loss of subcutaneous fat from the arms and legs, while in the face and neck, the deposition of fat is normal or excessive. MAD type B (OMIM 608612) is caused by mutations in the *ZMPSTE24* gene, which codes an endoprotease zinc metalloprotease ZMPSTE 24. This enzyme is necessary for the post-translational processing of prelamin A to mature lamin A. Mutations in the *ZMPSTE24* gene may cause the accumulation of the toxic farnesylated form of prelamin, resulting in the disruption of nuclear function in several tissues [198]. In patients with MAD type B, more generalized loss subcutaneous fat is observed [195,197,199,200,201]. Some patients may also have insulin resistance, hyperinsulinemia, diabetes mellitus, and hyperlipidemia [202].

SHORT syndrome (OMIM 269880)

SHORT syndrome (short stature, hyperextensibility of joins, ocular depression, Riger anomaly, teething delay) is associated with autosomal dominant inheritance mutations in the *PI3KR1* gene, which codes the PI3KR1 protein (regulatory p85α sub-unit) [13,203]. The p85α sub-unit is involved in the connection and stabilization of the p110 catalytic sub-unit and sub-units of the phosphatidylinositol 3-kinase (PI3K) protein that catalyzes the conversion of phosphatidyl phosphate 2 (PIP2) to PIP3, causing the activation of the AKT signaling pathway. The PI3K/AKT signaling pathway regulates cellular functions, such as differentiation, cell survival, and insulin’s biological action [13,203].

Werner syndrome (OMIM 277700)

Werner syndrome is caused by biallelic variants in the *WRN* gene that codes a dual DNA helicase/exonuclease enzyme. It is an autosomal recessive disorder [204]. Patients with this syndrome have a short stature due to the absence of the pubertal growth spurt, and are have prematurely aging features, with onset in the third decade, such as greying and thinning of the hair, bilateral cataracts, scleroderma-like skin changes, premature atherosclerosis, and early-onset cancers. These patients may suffer premature muscle loss from the limbs or sarcopenia. Diabetes mellitus, fatty liver disease and dyslipidemia are also commonly reported [205]. The congenital partial lipodystrophies are summarized in Table 3.

#### 2.7.3. Acquired Generalized Lipodystrophy

Acquired generalized lipodystrophy (AGL), also called Lawrence syndrome [206], is a rare syndrome, described in approximately 80 patients. It predominantly occurs in female patients [207,208]. This syndrome usually develops during childhood, adolescence or adulthood [194,209]. Its pathogenesis is unknown; however, as its possible etiology, some have suggested previous infection or autoimmune diseases, such as Hashimoto thyroiditis, rheumatoid arthritis, hemolytic anemia, chronic, active hepatitis, panniculitis, Sjögren syndrome, juvenile-onset dermatomyosities, systemic sclerosis, and systemic lupus erythematosus [209,210,211,212]. In 50% of one cohort of 40 patients with autoimmune- or panniculitis-associated AGL, anti-perilipin (anti-PLIN1) antibodies were detected. Therefore, a pathogenic role of anti-PLIN1 has been is suggested in the development of Lawrence syndrome [211,213,214]. It is suggested that autoantibodies and proinflammatory cytokines, such as TNF-α and IL-6, are involved in AGL, causing impaired fat uptake, the differentiation of adipocyte [57,212,215], adipogenesis [216], and the increased apoptosis of adipocytes/preadipocytes [217,218]. In most patients, the loss of fat begins in adolescence, and occurs variably over a period of weeks, months, or years. Low levels of leptin and very low levels of adipokines may be associated with metabolic complications, such as type 2 diabetes mellitus, hypertriglyceridemia, metabolic dysfunction-associated steatotic liver disease (MASCD), and severe insulin resistance [131]. Acquired generalized dystrophy is summarized in Table 4. 

#### 2.7.4. Acquired Partial Lipodystrophy 

Acquired partial lipodystrophy (OMIM 608709), also known as Barraquer–Simons syndrome, is one of the most commonly acquired lipodystrophy [207]. Its higher prevalence was detected in females, often after a fertility illness [219]. Approximately 250 cases have been reported [13]. This disorder begins in childhood or adolescence. Fat loss usually occurs over a period of months or years [220]. In patients with this dystrophy, the loss of adipose tissue from the face and upper trunk is detected, while in the rest of the body increased adiposity may be observed [221,222]. In these patients autoimmune disorders have been reported, such as dermatomyositis, hypothyroidism, pernicious anemia, rheumatoid arthritis, temporal arteritis, or mesangiocapillary glomerulonephritis [169,223,224]. Most patients with this syndrome have a circulating autoantibody, called C3 nephritic factor, and low complement component 3 (C3) levels [225,226]. C3 nephritic factor plays an important role, stabilizing C3 (the convertase enzyme) by increasing the half-life of the convertase due to the blockage of the degradation of C3. The excessive activation of C3 results [194]. In more than 20% of patients, the development of membraneproliferative glomerulonephritis (MPGN) is observed, which occurs on average 8 to 10 years after initial diagnosis [222]. The diagnosis of kidney impairment caused by MPGN is an important prognostic factor [220]. In patients, hyperinsulinemia may be detected, but without severe insulin resistance. The prevalence of diabetes mellitus in these patients (approximately 7%) is much lower when compared to other types of lipodystrophy [222]. The etiology of Barraquer–Simons syndrome remains unclear; however, the autoimmune-mediated destruction of adipocytes is suggested [13]. Acquired partial lipodystrophy is summarized in Table 5.

Lipodystrophy associated with HIV therapy

HIV-associated lipodystrophy syndrome (HALS) is the most common type of partial lipodystrophy. HALS is detected in approximately 40% of patients who are treated with highly active antiretroviral therapy (HAART), especially HIV-1 protease inhibitors and nucleoside analog reverse transcriptase inhibitors. Previously performed observations revealed that HALS depends on the duration of HAART treatment [112,227]. The above-mentioned drugs can induce the development of lipodystrophy, and these drugs may be the cause of lipodystrophy [220]. There are different mechanisms of these associations that have been suggested, such as the increased apoptosis of adipocytes, the inhibition of preadipocytes differentiation [228,229], and the suppression of adipogenesis [230]. These mechanisms also include the changed expression of adipogenic transcriptors, such as PPARγ, CCAAT/enhancer-binding protein-α (C/EBO-α), CCAAT/enhancer-binding protein-β (C/EBP-β), and sterol regulatory element-binding protein 1 (SREBP-1) [13,231]. Metabolic and clinical manifestations, as well as the severity of HALS, may be dependent on inflammation processes [232,233]. It is important to note that reduced subcutaneous adipose tissue is associated with a low concentration of leptin, while decreased levels of adipokine are associated with excess visceral fat [113,233].

Lipodystrophy associated with HIV therapy is summarized in Table 6.

#### 2.7.5. Localized Lipodystrophy Disorders

Localized lipodystrophy disorders are characterized by small amounts of subcutaneous fat loss from smaller areas of the body, and do not cause insulin resistance [114]. In these disorders, metabolic abnormalities are not observed, as the amount of fat loss is minimal. Localized lipodystrophy disorders include lipodystrophy caused by drug injections, lipodystrophy semicircularis, centrifugal lipodystrophy, and panniculitis-associated lipodystrophy [117]. Drug-induced lipodystrophy at the site of injection has been associated with insulin therapy. Given the use of purified human insulin, this disorder is rare now. Localized lipodystrophy may also be caused by glucocorticoids and antibiotics [169]. A rare cause of localized lipodystrophy is lipodystrophia centrifugalis abdominalis infantilis. In this lipodystrophy disorder, a centrifugal loss of subcutaneous fat in the abdomen is observed with erythematous and scaly changes at the periphery. It usually occurs before the age of three years. As a related abnormality, we might mention localized lipodystrophy, i.e., a lack of fat in small areas of trunk or parts of limbs [169,220]. Localized lipodystrophy disorders are summarized in Table 7.

#### 2.7.6. Other Complex Syndromes of Severe Insulin Resistance

Subcutaneous insulin resistance syndrome

Subcutaneous insulin resistance syndrome is a rare disorder, associated with resistance to the action of subcutaneous insulin, while patients maintain sensitivity to intravenous insulin. This disorder is caused by increased insulin-degrading enzyme activity in the subcutaneous tissue [234,235,236].

Alström syndrome

Alström syndrome (OMIM 203800) was first described in 1959 by Carl-Henry Alström [237]. It is caused by a mutation in the *ALMS1* gene. The ALMS1 protein is detected in primary cilia within the centrosomes and the basal bodies. It is suggested that the complications of the disease are due to dysfunctional cilia with complications arising early in life. The symptoms of the syndrome usually first arise in infancy, and further develop during childhood and later in life. It is characterized by the progressive development of multi-organ pathology [238]. In patients with Alström syndrome, endocrine, cardiac renal and hepatic complications may be observed, as well as complications with vision and hearing [239,240,241]. Patients with the Alström syndrome rarely exceed the age of 50. It is inherited as an autosomal recessive genetic disorder. Its prevalence is estimated to range from 1 in 500,000 to 1 in 1,000,000, and approximately 1200 cases of syndrome have been described [239]. Currently, there are no specific therapies for Alström syndrome.

Bloom syndrome

Bloom syndrome (OMIM 210900) was first described in 1954 by David Bloom [242]. Bloom syndrome, also called Bloom–Torre–Machacek syndrome, or congenital telangiectacic erythema, is caused by a mutation in the *BLM* gene, causing the formation of an abnormal DNA helicase protein. It is an extremely uncommon disorder, with only 281 patients described as of 2018 [243]. In patients features are observed such as growth retardation, photosensitive skin, a compromised immune system, insulin resistance, and a high predisposition to cancer [244,245]. The mean age of death is 26 years of age, most commonly caused by malignancies [243].

Microcephalic osteodysplastic primordial dwarfism type II

Microcephalic osteodysplastic primordial dwarfism type II (MOPD II) (OMIM 210720) was first described in 1982 by Majewski et al. [246]. MOPD II is caused by biallelic mutations in the pericentrin *PCNT* gene [247]. It is inherited as an autosomal recessive disorder [248]. In patients, features are detected of severe pre- and post-natal growth failure with microcephaly, characteristic facial features, skeletal dysplasia, abnormal dentition, insulin resistance and truncal obesity [249,250]. The mortality of patients with MOPD II predominantly occurs in adulthood. It is a rare disease with just over 150 cases described in the literature [249,251].

The characteristics of other syndromes related to severe insulin resistance are presented in Table 8.

For more details on insulin resistance syndromes, see [13,117,162,220,252,253].

## 3. Therapy

Severe insulin resistance syndromes may be caused by several factors, both genetic and environmental. The nature of SIRS is heterogeneous; therefore, the treatment options are different and have variable effects depending on the type of insulin resistance and the individual presentation of the disorder. The treatment of a severe resistance syndrome aims to ameliorate both the metabolic disturbances and pathophysiological changes. There are two treatment strategies: non-pharmacological treatment and pharmacological therapy.

### 3.1. Non-Pharmacological Treatment

Non-pharmacological treatment is associated with lifestyle modifications, such as physical activity and diet. Changes in lifestyle should start as early as possible in association with pharmaceutical treatment.

Increased physical activity is recommended in the primary management stage for several insulin resistance syndromes. Exercise stimulates the uptake of glucose and the synthesis of glycogen in skeletal muscle. In patients with diabetes mellitus, physical activity may improve insulin sensitivity and glucose metabolism [254]. Early postoperative mobilization minimalizes postoperative insulin resistance [255]. It was found that in patients with congenital lipodystrophies and with HALS, a combination of aerobic and resistance training improves cholesterol and triglyceride levels, as well as body composition [127]. The performed observations also revealed in these patients that resistance training alone increases total lean mass, increases HDL cholesterol levels, and improves peripheral insulin sensitivity [256,257]. Increased physical activity may be beneficial in preventing and improving lipodystrophy [258] and augments pharmacological therapy [259]. It is suggested that 30 minutes of daily moderate-intensity physical activity is beneficial [13]. Sedentary time should be reduced, and screen time should be reduced to less than two hours a day [260]. On the other hand, in patients with severe hepatosplenomegaly and CGL who have lytic bone lesions, or with cardiomyopathy, some kinds of exercise should be avoided [115,147]. Exercise is also not recommended in low-weight patients with primary lipoatrophy, because overly intense physical activity may exacerbate fat loss [261]. Physical activity should be associated with dietary intervention.

Dietary guidelines for patients with lipodystrophy have not been established. Caloric-restricted diets in these patients are very important, as they are typically hyperphagic [13]. These patients also have an elevated risk for cardiovascular diseases and diabetes. Therefore, for these patients, the American Heart Association recommends that less than 30% of daily calories come from fat [262], and according to the recommendations of the American Diabetes Association, carbohydrate and monounsaturated fat should provide 60–70% of daily calories [263]. Observations performed on women with PCOS revealed that these women tend to be high in carbohydrates and fat [264]. It is suggested that for a reduction in insulin resistance and improved body composition, the best option may be calorie-restricted diets [265]. The Mediterranean diet consists of a plentiful intake of vegetables, fruits, seafood, legumes, nuts, whole-grain cereals and olive oil. The results obtained in the studies performed show that the Mediterranean diet combined with a low-carbohydrate regimen improves endocrine disorders in overweight patients with PCOS [266,267]. There are limited data published on the Mediterranean diet and lipodystrophies. It was found that this diet may benefit patients with HALS [268,269]. It was observed that in the case of acquired total lipodystrophy, serum triglyceride levels and insulin resistance may be improved. This effect may be obtained by the substitution of long-chain fatty acids with medium-chain or n-3 polyunsaturated fatty acids [270].

### 3.2. Pharmacologic Treatment

Currently, there is a lack of guidelines that describe the best treatment of patients with SIRS [271], as well as a lack of medications approved for treating insulin resistance [272]. Therefore, as mentioned above, changes of lifestyle play an important role in the management of IR in addition to alternative medications. It has also been suggested that the use of pharmacologic medications, such as metformin, sulfonylureas, sodium-glucose transporter 2 (SGLT2) inhibitors, dipeptidyl peptidase 4 (DPP-4) inhibitors, α-glucosidase, and glucagon-like peptide 1 (GLP1) receptor agonists or insulin, may be a useful second step of therapy [273] (Table 9).

Metformin, a biguanide derivative, is recommended as the initial pharmacological option for adolescents with type 2 diabetes mellitus, in association with changed lifestyle. It reduces the hepatic synthesis of glucose by decreasing gluconeogenesis and stimulates peripheral glucose uptake. Metformin is involved in weight loss due to its anorexic effect [274]. In patients with lipodystrophies, metformin improves insulin sensitivity, and in patients with HALS, it may improve fat distribution [275]. Metformin improves insulin sensitivity by increasing insulin receptor tyrosine kinase activity, enhances the synthesis of glycogen and stimulates the translocation of glucose transporter GLUT4 [276,277]. It stimulates the re-estrification of free fatty acids and inhibits lipolysis, resulting in increased insulin sensitivity due to reduced lipotoxicity in adipose tissue [278]. This drug is widely used in treating hereditary severe insulin resistance syndrome, but it is not always effective. It was found that metformin plays a beneficial role in patients with PCOS. The results obtained in insulin-resistant human and animal studies show that using an insulin sensitizer improves insulin sensitivity, alleviates metabolic disorders and ameliorates polycystic symptoms [267]. The use of metformin may be limited by gastrointestinal side-effects, which may cause the discontinuation of therapy [271].

Thiazolidinediones (TZDs), true insulin sensitizers, particularly pioglitazone, improve insulin sensitivity, causing the activation of PPAR-γ and increasing the expression of genes involved in adipogenesis. However, TZDs do not improve lipoatrophy in patients with FPLD2, but may be efficacious in patients with HALS. The treatment of patients with HALS with pioglitazone for 12 months improves blood pressure, lipid profile, and insulin resistance in these patients [279]. In women with PCOS, TZDs improve abnormal glucose tolerance, insulin resistance and dyslipidemia [280,281]. There has also been a case of type A insulin resistance syndrome being successfully treated with pioglitazone and flutamide for 5 years. In patients with TAIRS, TZD attenuated insulin resistance [37].

Insulin-like growth factor-1 (IGF-1) and insulin mediate the effects of similar tyrosine kinase receptors. They can interchangeably activate alternate kinase receptors. IGF-1 may play a role as a therapeutic agent against insulin resistance. The performed observations reveal that recombinant human IGF-1 (rhIGF-1) improves metabolic control in severe insulin resistance syndrome caused by mutations in the *INSR* gene. It also increases life span in patients with Donohue syndrome [282,283]. RhIGF-1 regulates glucose homeostasis due to reduced gluconeogenesis and increased glucose uptake in peripheral tissues [284]. Unfortunately, the available publications only describe single-case reports, or include few patients [13]. Side effects have prevented its wider use.

Glucagon-like peptide 1 (GLP-1) receptor agonists (GLP-1RAs), such as liraglutide, semaglutide, dulaglutide, and exenatide, mimic the incretin secreted by the distal small intestine. GLP-1 is an intestinal hormone that stimulates insulin secretion, activating the GLP-1 receptor, which is expressed on islet β-cells. GLP-1RAs also bind to insulin receptors in the pancreatic β-cells, increasing the release of insulin and inhibiting glucagon secretion in the presence of elevated blood glucose [285]. These agonists, suppressing the inflammatory response of macrophages, inhibit insulin resistance [286].

The incretin hormones GLP-1 and glucose-dependent insulinotropic polypeptide (GIP) cause up to 65% of postprandial insulin secretion. Tirzepatide (TZP) is a new antidiabetic drug that may have a potential metabolic activity in the treatment of T2DM and IR. It is a dual GIP/GLP-1 receptor agonist. This drug is a synthetic linear polypeptide that binds to albumin, entailing a possibly week dose. TZP contains 39 amino acids conjugated with the C20 fatty acids molecule. It binds both receptors GIP (GIPR) and GLP-1R with high affinity compared to native GIP for GIPR, and is approximately five times weaker in comparison with native GLP-1 for GLP-1R [287,288]. Animal studies performed on wild mice and transgenic mice lacking GIPR or GLP-1R revealed that TZP is specific to GIPR and GLP-1R and active against both incretin receptors [289]. Results have also been obtained in human studies. The clinical efficacy, safety, and tolerance of TZP have been assessed in phase 1, 2, and 3 clinical trials. The results obtained in a phase 1 trial [289,290,291], in phase 2 clinical trials [292,293,294], and in phase 3 clinical trials [295] show a beneficial effect of TZP on patients with T2DM. TZP reduces glycemic levels, improves insulin sensitivity and β-cell functions, reduces body weight, and improves lipid metabolism, all of which are disturbances associated with T2DM and IR [296]. Retatruide (RETA) is a novel triple agonist of the GIP, GLP-1, and glucagon (GCG) receptors. It is a single protein conjugated to a fatty acid moiety that activates the mentioned receptors. It is less potent in comparison to endogenous ligands of human GCG (0.3 times as active) and GLP-1 (0.4 times as active). On the other hand, it is more potent as the human GIP receptor (by a factor 8.9) [297]. Its mechanism of action is associated with a synergistic interaction among these receptors, causing increased insulin secretion, improved glucose homeostasis, and refined appetite modulation. In patients with diseases and metabolic disorders treated with RETA, beneficial effects were observed. The treatment of patients with type 2 diabetes mellitus improves lipid profile and decreases blood pressure [298], reveals clinically meaningful improvements in glycemic control, and reduces body weight [299,300,301]. It may play a role in mitigating cardiovascular risk factors and metabolic dysfunction-associated steatotic liver disease [300]. Nonalcoholic fatty liver disease, now termed metabolic dysfunction-associated steatotic liver disease (MASLD), is a chronic liver disease. It was found that obesity is associated with the growing burden of MASLD. An important pathophysiological driver of MASLD is insulin resistance. IR in adipocytes dysregulates lipolysis, resulting in the excessive delivery of fatty acids to the liver. The results obtained in a randomized phase 2a trial revealed that the treatment of obese people with retatruide decreases body weight and abdominal fat, and improves insulin sensitivity and lipid metabolism [302].

Sulfonylureas, such as glimepiride, promote the activation of the insulin receptor, causing an increase in the amount of glucose transporters, with a resultant increase in insulin sensitivity and improved insulin resistance [273].

Dipeptidyl Peptidase-4 (DPP-4) inhibitors, such as gencigliptin and saxagliptin et al., are used worldwide as therapies for T2DM, and may improve insulin sensitivity [303]. They prolong the activity of endogenous GLP-1 and insulinotropic polypeptide. These inhibitors may offer an alternative to GLP-1 receptor agonists for the management of patients with severe insulin resistance [271]. DPP-4 inhibitors are well tolerated by patients, but their role in the management of patients with severe insulin resistance is questionable [304,305].

Sodium-Glucose Cotransporter 2 (SGLT2) inhibitors, such as canagliflozin, dapagliflozin and ampagliflozin, are used in combination with diet and exercise to lower blood glucose levels and to stimulate insulin secretion in diabetic patients [306,307]. These inhibitors increase the excretion of urinary glucose, reducing blood glucose levels. Observations have also revealed that the mentioned inhibitors reduce insulin resistance, protect the function of pancreatic β-cells, and reduce body weight [271,308,309].

Metrelepin is a recombinant human leptin approved to treat congenital generalized lipodystrophies [310]; it improves glycemic control and alleviates hypertriglyceridemia [311].

Immunosuppressants, such as rituximab, are used in patients with type 2 insulin resistance. These patients have also been treated with intravenous immunoglobulin. Unfortunately, different results have been derived [63,312]. For example, in seven patients with TBIRS treated with rituximab, remission has been observed. But after eight months, this therapy was stopped, and the status of insulin resistance remained unclear [13,79].

Comparisons and characteristics of pharmacological therapies for SIRS are presented in Table 9.

## 4. Summary

This review has described the pathogenesis and features of potential therapeutic interventions for SIRS. There are several medications available to treat patients with diabetes mellitus, but only a few studies have been performed on SIRS. The majority of patients with severe insulin resistance syndrome are treated with metformin. Recent investigations of SIRS are timely but limited, and so further research is needed. Also, educational initiatives regarding lipodystrophy may help improve diagnosis, and also may be important for improving the treatment and management of SIRS.

## Figures and Tables

**Table 1 ijms-26-05669-t001:** Severe insulin resistance syndromes due to mutations in the insulin receptor gene and/or disturbances in the insulin signaling pathway.

Syndrome	Symptoms	Pathogenesis
Type A Insulin resistance (TAIRS) OMIM 610549 Prevalence— 1 in 100,000, female predominance.	Symptoms present in adolescence or adulthood. Hyperandrogenism, severe insulin resistance, acanthosis nigricans, hirsutism, oligomenorrhea, polycystic ovaries, acromegaloid features. Occasionally not obese.	Mutations in insulin receptor gene and impaired insulin signaling pathway. Not life-threatening, patients can live beyond middle age. Inheritance—autosomal dominant or autosomal recessive.
Type B Insulin resistance (TBIRS) ORPHA 2298. Rare, no prevalence reported.	Symptoms are detected in adulthood. Acanthosis nigricans, refractory hyperglycemia, severe insulin resistance, markedly enlarged ovaries, hyperandrogenism, features associated with autoimmunity.	Autoantibody against insulin receptor. Frequently associated with autoimmune disorders. Mortality over 50%. Inheritance—not known.
Type C Insulin resistance. A variant of type A, the HAIR-AN syndrome. It affects 2–3% of androgen excess patients.	A variant of type A insulin resistance.	Reduced activity of insulin receptor kinase Generally not life-threatening. Inheritance—not known.
Donohue syndrome (leprechaunism) OMIM 246200. Prevalence— 1 in 1,000,000 live births.	Congenital onset, delayed vertical growth, hyperinsulinemia, acanthosis nigricans, postprandial hyperglycemia, hypotonia, proptosis, infraorbital folds, cardiomegaly, hepatosplenomegaly, hypertrophy of kidneys and ovaries.	Mutations in insulin receptor gene, caused disrupted binding of insulin to its receptor. Death usually during infancy. Inheritance—autosomal recessive.
Rabson–Mendenhall syndrome OMIM 262190. Prevalence—its exact prevalence has not been assessed; however, it is suggested that its prevalence is less than 1 in 1,000,000 live births.	Congenital onset, growth retardation, wasting of muscles, lack of subcutaneous fat, hyperinsulinemia, polycystic ovaries, acanthosis nigricans, insulin-resistant diabetes, dental abnormality, pineal hyperplasia.	Mutations in insulin receptor gene and impaired insulin signal transduction. Patients survive early childhood, but they have a significantly reduced life expectancy and may die during adolescence or early adulthood. Patients usually survive beyond one year of age. Inheritance—autosomal recessive.

**Table 2 ijms-26-05669-t002:** Characteristics of congenital generalized lipodystrophies.

LIPODYSTROPHIES		
Congenital Generalized Lipodystrophies (CGL)—Seip–Berrardinelli Syndrome (OMIM 269700)		
Syndrome	Symptoms	Pathogenesis
CGL1 OMIM 608594 It has been identified in at least 150 cases.	Lytic bone lesions, cardiomyopathy. The normal adipose tissue deposition is limited to mechanical fat, as well as the mouth, tongue, scalp and perineum.	Mutations in *AGPAT2* gene. Deficiency of AGPAT2 impairs signaling pathways of PI3K/AKT and PPARγ, causing affected adipogenesis and reducing the levels of stored triglycerides. Inheritance—autosomal recessive.
CGL2 OMIM 269700 It has been identified in at least 167 patients.	Absence of body fat with loss of both metabolically active fat and mechanical fat. Since birth, mild mental retardation, dyslipidemia, cardiomyopathy.	Mutations in the *BSCL2* gene that codes seipin. It is more severe than CGL1 and features a higher incidence of premature death. Some variants of the *BSCL2* gene are fatal, due to encephalopathy in early childhood. Inheritance—autosomal recessive.
CGL3 OMIM 612526 No prevalence reported.	Impaired function of caveolin may cause lipodystrophy. Short stature, megaesophagus, dyslipidemia, generalized lipodystrophy from birth.	Mutations in the *CAV1* gene that codes caveolin-1. Inheritance—autosomal recessive.
CGL4 Only 21 patients with CGL4 have been reported.	Moderate lipodystrophy, hypertriglyceridemia, cardiomyopathy, cardiac fibrosis.	Mutations in the *PTRF* gene that codes cavin, a polymerase I and transcript release factor, QT interval prolongation, atlantoaxial instability, gastrointestinal disorders. It can be associated with sudden death. Inheritance—autosomal recessive.

**Table 3 ijms-26-05669-t003:** Characteristics of congenital partial lipodystrophies.

LIPODYSTROPHIES		
Congenital Partial Lipodystrophies		
Familial Partial Lipodystrophies (FPLD)		
Syndrome	Symptoms	Pathogenesis
FPLD1 Köbberling syndrome OMIM 608600 It is predominantly diagnosed in females.	The characteristic clinical features are not completely defined. In patients, the loss of adipose tissue in the extremities is observed, with normal adipose tissue elsewhere. Most affected individuals have diabetes and hypertriglyceridemia that may cause acute pancreatitis.	Little is known about pathophysiological mechanisms. The genetic basis is unknown. A polygenic inheritance is suggested in some patients.
FPLD2 Dunningan syndrome OMIM 151660	Loss of subcutaneous tissue from the extremities. Some patients may have acanthosis nigricans, hirsutism, and ovarian hyperandrogenism. Prior to the age of 20, diabetes and hepatic steatosis may develop.	Mutations in *LMNA* gene that codes lamins A and C. The impaired protein may disrupt the interaction with chromatin or other nuclear lamin proteins, causing apoptosis and the premature death of adipocytes. It may be inherited as an X-linked dominant or autosomal dominant trait.
FPLD3 OMIM 604367	Lipoatrophy appears in adulthood. The symptoms are similar to those of FPLD2; however, fat deposition in these patients in the head and neck may be normal. FPLD3 patients have a greater prevalence of hypertriglyceridemia and diabetes.	Mutations in the *PPARG* gene that participates in adipocyte differentiation and its function. These mutations impair adipocyte differentiation and adipogenesis. Inheritance—autosomal dominant.
FPLD4 OMIM 613877 It was described in four families.	Loss of subcutaneous fat in the extremities. The syndrome may be associated with severe insulin resistance, diabetes, hypertriglyceridemia, and hepatic steatosis.	Mutations in the *PLIN1* gene that codes perilipin-1. It is involved in lipid storage and lipolysis. Inheritance—autosomal dominant.
FPLD5 OMIM 615238	Partial lipodystrophy, acanthosis nigricans, severe insulin resistance, diabetes, hepatic steatosis, peripheral lipoatrophy, visceral adiposity.	Mutations in the *CIDEC* gene. These mutations impair adipocyte differentiation and lipid droplet accumulation. Inheritance—autosomal recessive.
FPLD6 OMIM 615980	Distal lipoatrophy, visceral adiposity, and muscular dystrophy.	Mutations in the *LIPE* gene that codes lipase E. It is involved in the regulation of adipocyte function, and the homeostasis of lipid and glucose. These mutations impair lipolysis, resulting in lipomatosis and partial fat loss. Inheritance—autosomal recessive.
*AKT2*-linked lipodystrophy	Insulin resistance, moderate lipodystrophy.	Mutations in the *AKT2* gene that codes protein kinase B, a serine/threonine protein kinase. Mutations impair insulin signaling and adipocyte differentiation. Inheritance—autosomal dominant.
*CAV1*-associated lipodystrophy OMIM 606721. Only two cases were reported.	Subcutaneous fat loss in the face and upper body, micrognathia, congenital cataracts, diabetes mellitus, hypertriglyceridemia, recurrent pancreatitis.	Mutations in the *CAV1* gene that codes caveolin-1, the membrane protein associated with cell metabolism, cell migration, and cell signaling. Inheritance—autosomal dominant.
Mandibuloacral dysplasia type A OMIM 249370	Loss of subcutaneous fat from the arms and legs. Skeletal abnormalities, mandibular and clavicular hypoplasia, delayed dentition, progeroid features.	Mutations in the *LMNA* gene that codes lamin A and C, nuclear lamin proteins. Impaired nuclear functions, causing premature cell death in adipose and skeletal tissue. Inheritance—autosomal recessive.
Mandibuloacral dysplasia type B OMIM 608612	Generalized loss of fat, mandibular and clavicular hypoplasia, acrosteolysis, delayed dentition, premature renal failure, progeroid features.	Mutations in the *ZMPSTE24* gene that codes zinc and metalloprotease ZMPSTE24, involved in the processing of prelamin A to mature lamin A. Impaired gene may cause the accumulation of the toxic farnesylated form of lamin, resulting in the impaired nuclear function in several tissues. Inheritance—autosomal recessive.
SHORT syndrome OMIM 269880	Short stature, hyperextensibility of joins, ocular depression, Riger anomaly, teething delay.	Mutations in the *PI3KR1* gene that codes PI3KR1 protein (regulatory p85α subunit). The p85α subunit is associated with the connection and stabilization of the p110 catalytic subunit of the phosphatidyl-inositol 3-kinase (PI3K) protein, which catalyzes the conversion of phosphatidyl phosphate 2 (PIP2) to PIP3, causing the activation of the AKT signaling pathway. The PI3K/AKT signaling pathway regulates cellular functions, such as differentiation, cell survival, and insulin’s biological action. Inheritance—autosomal dominant.
Werner syndrome OMIM 277700	Short stature, due to the absence of the pubertal growth spurt, premature aging of features, with onset into the third decade, such as greying and thinning of the hair, bilateral cataracts, scleroderma-like skin changes, premature atherosclerosis, and development of early-onset cancers. These patients may suffer premature muscle loss from the limbs or sarcopenia. Diabetes mellitus, fatty liver disease and dyslipidemia are also commonly reported.	Biallelic variants in the *WRN* gene that codes a dual DNA helicase/exonuclease enzyme. Inheritance—autosomal recessive.

**Table 4 ijms-26-05669-t004:** Characteristic of acquired generalized lipodystrophy.

LIPODYSTROPHY		
Acquired Generalized Lipodystrophy (AGL)—Lawrence Syndrome		
Syndrome	Symptoms	Pathogenesis
Lawrence syndrome It was described in approximately 80 patients. Predominantly occurs in females.	It usually develops during childhood, adolescence or adulthood. In most patients, the loss of fat begins in adolescence, and occurs variably over a period of weeks, months, or years. Low levels of leptin and very low levels of adipokines may be associated with metabolic complications, such as type 2 diabetes mellitus, hypertriglyceridemia, metabolic dysfunction-associated steatotic liver disease (MASCD), and severe insulin resistance.	Its pathogenesis is unknown, although it is suggested that previous infection or autoimmune diseases, such as Hashimoto thyroiditis, rheumatoid arthritis, hemolytic anemia, chronic, active hepatitis, panniculitis, Sjögren syndrome, juvenile-onset dermatomyosities, systemic sclerosis, and systemic lupus erythematosus, may be relevant. In 50% of patients with autoimmune- or panniculitis-associated AGL, anti-perilipin (anti-PLIN1) antibodies were detected. Therefore, it is suggested that a pathogenic role of anti-PLIN1 is involved in the development of the Lawrence syndrome. It is suggested that autoantibodies and proinflammatory cytokines, such as TNF-α and IL-6, are involved in AGL, causing impaired fat uptake, the differentiation of adipocyte, adipogenesis, and the increased apoptosis of adipocytes/pre-adipocytes.

**Table 5 ijms-26-05669-t005:** Characteristics of acquired partial lipodystrophy.

LIPODYSTROPHIES		
Acquired Partial Lipodystrophy		
Syndrome	Symptoms	Pathogenesis
Barraquer–Simons syndrome OMIM 608709 One of the most commonly acquired types of lipodystrophy, approximately 250 cases have been reported. Its higher prevalence was detected in females, often after a fertility illness.	It begins in childhood or adolescence. Fat loss usually occurs over a period of months or years. A loss of adipose tissue from the face and upper trunk is detected, while in the rest of the body increased adiposity may be observed. In patients, autoimmune disorders have been reported, such as dermatomyositis, hypothyroidism, pernicious anemia, rheumatoid arthritis, temporal arteritis, or mesangiocapillary glomerulonephritis. Most patients with this syndrome have a circulating autoantibody called C3 nephritic factor and low complement component 3 (C3) levels. C3 nephritic factor plays an important role, stabilizing the C3 convertase enzyme by increasing the half-life of the convertase by blocking the degradation of C3. A result is the excessive activation of C3. In some patients the development of membrane-proliferative glomerulonephritis (MPGN) is observed, which occurs on average 8 to 10 years after initial diagnosis. In patients hyperinsulinemia may be detected, but without severe insulin resistance. The prevalence of diabetes mellitus in these patients is much lower as compared to other types of lipodystrophy.	The etiology of Barraquer–Simons syndrome remains unclear; however, the autoimmune-mediated destruction of adipocytes is suggested.

**Table 6 ijms-26-05669-t006:** Characteristic of lipodystrophy associated with HIV therapy.

LIPODYSTROPHY		
Lipodystrophy Associated with HIV Therapy		
Syndrome	Symptoms	Pathogenesis
HIV-associated lipodystrophy syndrome (HALS). It is the most common type of partial lipodystrophy.	HALS is detected in approximately 40% of patients, who are treated with highly active antiretroviral therapy (HAART), especially HIV-1 protease inhibitors and nucleoside analogue reverse transcriptase inhibitors. HALS depends on the duration of HAART treatment. The drugs used can develop lipodystrophy, and these drugs may be the cause of lipodystrophy. Reduced subcutaneous adipose tissue is associated with a low concentration of leptin, while decreased levels of adipokine are associated with excess visceral fat.	There are different mechanisms of these associations suggested, such as the increased apoptosis of adipocytes, the inhibition of preadipocytes differentiation, and the suppression of adipogenesis. These mechanisms also include the changed expression of adipogenic transcriptors, such as PPARγ, CCAAT/enhancer-binding protein-α (C/EBO-α), CCAAT/enhancer-binding protein-β (C/EBP-β), and sterol regulatory element-binding protein 1 (SREBP-1). Metabolic and clinical manifestations, as well as the severity of HALS, may be dependent on inflammation processes.

**Table 7 ijms-26-05669-t007:** Characteristics of localized lipodystrophy disorders.

LIPODYSTROPHY		
Localized Lipodystrophy Disorders		
Syndrome	Symptoms	Pathogenesis
Localized lipodystrophy disorders	These disorders are characterized by small amounts of subcutaneous fat loss from small areas of the body, and do not cause insulin resistance. There are no observed metabolic abnormalities, as the amount of fat loss is minimal. A rare cause of localized lipodystrophy is lipodystrophia centrifugalis abdominalis infantilis. In this lipodystrophy disorder, a centrifugal loss of subcutaneous fat is observed in the abdomen with erythematous and scaly changes at the periphery. It usually occurs before the age of three years. As a related abnormality, we may cite localized lipodystrophy, i.e., a lack of fat in small areas of trunks or parts of limbs.	Localized lipodystrophy disorders include lipodystrophy caused by drug injections, lipodystrophy semicircularis, centrifugal lipodystrophy, and panniculitis-associated lipodystrophy. Drug-induced lipodystrophy at the site of injection is associated with insulin therapy. The use of purified human insulin means that this disorder is rare now. Localized lipodystrophy may also be caused by glucocorticoids and antibiotics.

**Table 8 ijms-26-05669-t008:** Characteristics of other complex syndromes related to severe insulin resistance.

Other Complex Syndromes of Severe Insulin Resistance		
Syndrome	Symptoms	Pathogenesis
Subcutaneous insulin resistance syndrome	Resistance to the action of subcutaneous insulin.	This syndrome occurs due to increased insulin-degrading enzyme activity in the subcutaneous tissue.
Alström syndrome	Endocrine, cardiac renal and hepatic complications, and complications with vision and hearing, may be observed.	It is caused by a mutation in the *ALMS1* gene.
Bloom syndrome	In patientsgrowth retardation, photosensitive skin, compromised immune system, insulin resistance, and a high predisposition of cancer are observed. The mean age of death is 26, most commonly caused by malignancies.	It is caused by a mutation in the *BLM* gene.
Microcephalic osteodysplastic primordial dwarfism type II	Pre- and post-natal growth failure with microcephaly, characteristic facial features, skeletal dysplasia, abnormal dentition, insulin resistance and truncal obesity are observed.	This syndrome is caused by a mutation in the pericentrin *PCNT* gene.

**Table 9 ijms-26-05669-t009:** Characteristics and comparison of medications used in pharmacologic treatment.

Medications	Mechanism of Action and Effects
Metformin	Decreases gluconeogenesis, stimulates peripheral glucose uptake, and causes weight loss. In patients with lipodystrophies, especially with HALS, it improves insulin sensitivity, increases the activation of insulin receptor tyrosine kinase, and enhances the synthesis of glycogen.
IGF-1	Improves metabolic control in SIRS caused by mutations in the gene, increases life span, regulates glucose homeostasis, reduces hepatic gluconeogenesis and increases glucose uptake in peripheral tissues.
GLP-1RAs	Stimulates insulin secretion, activates GLP-1 receptor, increases the secretion of insulin, inhibits the secretion of glucagon and inhibits insulin resistance.
Sulfonylureas	Activates insulin receptors, increases the amount of glucose transporters, increases insulin sensitivity and improves insulin resistance.
DPP-4 inhibitors	Improves insulin sensitivity, stimulates the activity of endogenous GLP-1 and insulinotropic polypeptide.
SGLT2 inhibitors	Decrease blood glucose levels, stimulates insulin secretion and increases the excretion of urinary glucose, reducing blood glucose levels. Reduce insulin resistance and body weight.
Metreleptin	This is recommended for treating congenital generalized lipodystrophies. It improves glycemic control and alleviates hypertriglyceridemia.
Immunosuppressant	There are different and controversial results.

## Data Availability

Not applicable.

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
