# Peer review of "Severe Insulin Resistance Syndromes: Clinical Spectrum and Management"

_ijms, 2025, doi:10.3390/ijms26125669_

Round 1
Reviewer 1 Report
Comments and Suggestions for Authors
This is a comprehensive overview of severe insulin resistance syndromes. The text of manuscript is very long but presentation is detailed. The authors must improve English and explain all abbreviations at first application in the text. In my opinion Table 1 should be divided in more ones.
New therapies like tirzepatide (dual GIP and GLP-1RA) and retatrutide (triple) should be discussed.
Other remarks are marked in the file

Medical English should be improved by native speaker
Reviewer 2 Report
Comments and Suggestions for Authors
The review by Pliszka and Szablewski aims to provide a wide-ranging summary of SIRS, covering their pathophysiology, complications, prognosis and classification. It also includes a final paragraph on the various treatment options available.
Overall, this review offers an up-to-date, comprehensive overview of SIRS. However, several paragraphs need to be revised for grammar (e.g., 'cells fail to adequately respond to insulin').
Furthermore, the introduction is rather long, and some sentences are redundant. For example, ‘There are also some cases, that organs may develop resistance to insulin, causing affected normal insulin action and cells sensitive to insulin, do not respond to its levels in the body'.
'Severe insulin resistance syndromes' should always be abbreviated to 'SIRS' in the text, not just in the therapy and summary sections.
I suggest including a description of the other SIRS mentioned in Section 2.7.6 instead of only mentioning references.
Finally, although Table 1 is intended to summarise SIRS characteristics, it should be revised and streamlined, and columns should be added if necessary to include more information
